# Investigation of Outbreaks of Extended-Spectrum Beta-Lactamase-Producing Klebsiella Pneumoniae in Three Neonatal Intensive Care Units Using Whole Genome Sequencing

**DOI:** 10.3390/antibiotics9100705

**Published:** 2020-10-16

**Authors:** Sammy Frenk, Nadya Rakovitsky, Elizabeth Temkin, Vered Schechner, Regev Cohen, Bat Sheva Kloyzner, Mitchell J. Schwaber, Ester Solter, Shoshana Cohen, Sarit Stepansky, Yehuda Carmeli

**Affiliations:** 1National Institute for Antibiotic Resistance and Infection Control, Ministry of Health, Tel Aviv 64239, Israel; nadyarak@tlvmc.gov.il (N.R.); lizt@tlvmc.gov.il (E.T.); mitchells@tlvmc.gov.il (M.J.S.); esterso@tlvmc.gov.il (E.S.); yehudac@tlvmc.gov.il (Y.C.); 2Division of Epidemiology and Preventive Medicine, Tel Aviv Sourasky Medical Center, Tel Aviv 64239, Israel; vereds@tlvmc.gov.il (V.S.); saritst@tlvmc.gov.il (S.S.); 3Sackler Faculty of Medicine, Tel Aviv University, Tel Aviv 69978, Israel; 4Infectious Diseases Unit, Laniado Hospital, Netanya 42150, Israel; regevc@laniado.org.il (R.C.); shoshic@laniado.org.il (S.C.); 5Ruth and Bruce Rappaport Faculty of Medicine, Technion-Israel Institute of Technology, Haifa 3525433, Israel; 6Department of Epidemiology, Mayanei Hayeshua Medical Center, Bnei Brak 51544, Israel; kloyz@mhmc.co.il

**Keywords:** ESBL-KP, dual technology WGS, neonatal ICU, outbreak

## Abstract

Infections caused by extended-spectrum beta-lactamase-producing *Klebsiella pneumoniae* (ESBL-KP) are on a constant rise and are a noted cause of outbreaks in neonatal intensive care units (NICUs). We used whole genome sequencing (WGS) to investigate the epidemiology of consecutive and overlapping outbreaks caused by ESBL-KP in NICUs in three hospitals in close proximity. Clonality of 43 ESBL-KP isolates from 40 patients was determined by BOX-PCR. Short-read sequencing was performed on representative isolates from each clone. The dominant clones from each NICU were sequenced using long-read sequencing. Bioinformatics methods were used to define multilocus sequence type (MLST), analyze plasmid content, resistomes, and virulence factors. In each NICU, we found a unique dominant clone (ST985, ST37, and ST35), each belonging to a distinct sequence type (ST), as well as satellite clones. A satellite strain in NICU-2 (ST35) was the dominant strain in NICU-3, where it was isolated four weeks later, suggesting transmission. NICU-1- and NICU-2-dominant strains had *bla*_CTX-M-15_ carried on a similar transposable element (Tn3-IS*Ecp1*) but at different locations: on a plasmid and on the chromosome, respectively. We concluded that the overlapping ESBL-KP outbreaks were a combination of clonal transmission within NICUs, possible transposable element transmission between NICUs, and repeated importation of ESBL-KP from the community.

## 1. Introduction

Extended-spectrum beta-lactamase-producing *Enterobacterales* (ESBL-PE) are considered a “serious threat” by the US Centers for Disease Control [1] and a “critical priority” for antibiotic development by the World Health Organization [2]. A study of 890 US hospitals estimated that the incidence of infections caused by ESBL-PE increased by 53% from 2012 to 2017 [3]. While ESBL-PE were initially almost exclusively nosocomial pathogens, over the last two decades the epidemiology has shifted and ESBL-PE, including *K. pneumoniae*, are now carried and spread in the community as well [4].

ESBL-producing *K. pneumoniae* (ESBL-KP) have been a noted cause of outbreaks in neonatal intensive care units (NICUs) [5,6,7,8]. Strains involved in these outbreaks classically represent transmission of a single clone, spread by the hands of healthcare workers or by fomites [9]. Such clonal outbreaks may be spread between wards and institutions by patient transfers or by healthcare workers who work at more than one site [7]. In these scenarios, a lack of clonal relatedness among infected patients or affected wards does not necessarily rule out nosocomial spread, as ESBL genes may spread by plasmids or transposable elements [8]. On the other hand, with the rise of community carriage of ESBL-producing strains, what appear to be nosocomial outbreaks in the NICU may actually represent multiple introductions from the community, either by vertical transmission during delivery or by visiting family members who are ESBL-PE carriers. In these scenarios, it is expected that the strains and plasmids will be unrelated in contrast to cases of clonal spread.

Between July and September 2017, the National Center for Antibiotic Resistance and Infection Control in the Israeli Ministry of Health was notified independently by NICUs in three hospitals in a 20-km radius of outbreaks caused by ESBL-KP. In this study, we describe the molecular investigation of the consecutive (slightly overlapping) outbreaks using a methodological approach that included typing of all isolates and whole genome sequencing (WGS) of isolates representing clonal groups. We aimed to determine whether these outbreaks represented nosocomial transmission of a single clone or plasmid between all three institutions, separate nosocomial transmission within each institution, or unrelated events of ESBL importation from the community.

## 2. Results

Forty-three isolates from 40 unique patients in the three NICUs were included in this study. Four isolates from different anatomic sites came from one patient and were not included in the epidemiological timeline. All isolates were third-generation cephalosporin-resistant *K. pneumoniae* that were suspected ESBL producers. An epidemiological investigation was conducted without yielding conclusive results, which prompted a molecular analysis to answer the following questions.

### 2.1. Was This a Clonal Outbreak Involving All Three Hospitals?

BOX-PCR identified a distinct dominant clone in each of the three NICUs, as well as secondary and satellite (singleton) clones (Figure 1 and Appendix A). Although this finding suggests that the outbreaks were not clonally related, to further evaluate the possibility that the different clones did, in fact, share a common origin, we selected representative isolates from each clone for WGS. We found that each of the BOX-PCR clones belonged to a distinct ST-type (Figure 1). All sequence types belonged to the KpI phylogroup but were distant from each other (Figure 2). One satellite clone (Clone E, ST35) isolated from a patient in NICU-2 on 11 August 2017 was identified as the dominant clone in the outbreak in NICU-3, where it was first isolated four weeks later. The ST35 NICU-2 isolate was indistinguishable from the ST35 NICU-3 isolates by core-genome alignment: almost no dissimilarity distance was found between them (Appendix A). We believe that this represents transmission from NICU-2 to NICU-3 or a reoccurring introduction from a common reservoir. Thus, this was not a single clonal outbreak involving all three hospitals; rather, a suspected transmission event occurred between NICU-2 and NICU-3, leading to a clonal outbreak in NICU-3.

### 2.2. Were the Outbreaks a Result of Plasmid Dissemination Between Clones and NICUs?

All three dominant clones had two similar replication loci, IncFIB and IncFII, but the pMLST of their IncFII differed (Appendix A). Only the dominant clone in NICU-2 had IncFIA, and only the dominant clone in NICU-3 had a secondary IncFIB replication locus. Complete genome sequencing of representative isolates revealed a single plasmid in the NICU-1-dominant clone and two different plasmids in the NICU-2- and NICU-3-dominant clones. IncFIB(K)-IncFII (in NICU-1 and NICU-3) and IncFIB (in NICU-2) had similar operon composition and carried multiple antibiotic resistance genes, but had different gene arrangements and antibiotic resistance islands (ARI) (Figure 3). All three dominant clones carried several similar plasmid operons, including iron uptake (*fec*), copper (*pco*), arsenic (*ars*) and silver (*sil*) resistance. Surprisingly, the plasmids of the NICU-2-dominant clone did not carry a *bla*_CTX-M_ gene; instead, its *bla*_CTX-M-15_ was carried chromosomally. 

The ST35 satellite isolate in NICU-2 had the same plasmid profile as the ST35-dominant clone in NICU-3. None of the other satellite clones shared a plasmid profile with the dominant clones. In NICU-1, one satellite isolate (isolate 11, ST46) and the dominant clone had different plasmids but shared a similar ARI (*bla*_OXA-1_, *bla*_TEM-1_, *bla*_CTX-M-15_, *strB*, *strA* and *sul2*). This may represent ARI dissemination, resulting in a mosaic plasmid.

In summary, we found no plasmid dissemination across clones. The NICU-2-dominant clone’s plasmids did not carry a *bla*_CTX-M_ gene. The ST35 strain in NICU-2 and NICU-3 had the same plasmid profile.

### 2.3. Were the Outbreaks the Result of Dissemination of a Bla_CTX-M_ Transposable Element?

The *bla*_CTX-M-15_ gene was present in all NICU-1 isolates and in all NICU-2 isolates except for the ST35 isolate. All ST35 isolates carried a *bla*_CTX-M-14_ gene, while the two satellite clones in NICU-3 (ST268 and ST15) carried a *bla*_CTX-M-15_ gene. The *bla*_CTX-M-15_ gene in NICU-1 was carried on an IS*Ecp1* insertion sequence (IS) element. In the dominant clone, the IS was within a disrupted Tn3 transposon and flanked by antibiotic resistance genes (*bla*_TEM-1_, *bla*_OXA-1_, *strB*, *strA* and *sul2*) while in the satellite clones, other diverse flanking elements were found (Appendix A). In the NICU-2-dominant strain, *bla*_CTX-M-15_ was identified on the same IS*Ecp1* IS within the Tn3 transposon as in NICU-1. Interestingly, in the NICU-2-dominant clone, the transposome was carried on the chromosome and not on a plasmid. This IS*Ecp1*-*bla*_CTX-M-15_ fragment has previously been described on the chromosomes of isolates belonging to several ST types [10,11,12]. A comparison of these fragments is illustrated in Figure 4. The IS*Ecp1*-*bla*_CTX-M-15_ fragments from the dominant clones in NICU-1 and NICU-2 are more closely related to each other than to the fragments described in previous studies. In particular, the Tn3-like transposon is unique to the fragments from NICU-1 and NICU-2. The NICU-3-dominant clone had a *bla*_CTX-M-14_ gene adjacent to the transposable elements IS*Ecp1* (upstream) and IS5 (downstream).

In summary, the NICU-1- and NICU-2-dominant clones had *bla*_CTX-M-15_ carried on a similar transposable element (Tn3-IS*Ecp1*) but at different locations, i.e., on a plasmid in NICU-1 and on a chromosome in NICU-2. Because Tn3-IS*Ecp1*-*bla*_CTX-M-15_ is a common genetic complex, we cannot determine whether it was transmitted from NICU-1 to NICU-2 or introduced into both independently.

### 2.4. What May Have Contributed to the Success of the Dominant Clones?

*AMR genes.* All isolates were resistant to expanded-spectrum cephalosporins agents and were multi-drug resistant. Satellite clones generally had fewer AMR genes than the dominant clones: a mean of 9 vs. 8.6 (Appendix A). Examining specific resistance genes and gene families, we found that almost all clones had QNR genes and most had a *sul2* gene. The NICU-3-dominant clone had a *sul1* gene and the NICU-2-dominant clone had both sul1 and sul2 genes. The NICU-1-dominant clone carried 2 AAC genes (*aac(6′)-Ib* and *aac(3′)-IIa*). All study clones carried both a *bla*_CTX-M_ gene and an ESBL SHV gene. Thus, carrying genes from two ESBL families was not unique to dominant clones. In NICU-1, the dominant clone (ST985) and two satellite clones (ST13 and ST46) also carried a class D beta-lactamase, *bla*_OXA-1_. Because *bla*_OXA-1_ hydrolyzes broad-spectrum penicillins such as piperacillin and is not inhibited by tazobactam, it may confer selective advantage beyond ESBL carriage in settings where piperacillin/tazobactam is widely used. In sum, we found no significant difference in resistance potential between the dominant and satellite clones.

*Virulence genes.* Few virulence genes were identified in the study clones, but the dominant clone in NICU-1 and NICU-2 and one of the satellite strains in NICU-3 carried genes coding for yersiniabactin, a siderophore. Each clone had a different variant of yersiniabactin with different integrative conjugative elements (ICE) (Appendix A), suggesting independent events of gene acquisition. All dominant clones carried iron uptake (*fec*), copper resistance (*pco*), arsenic resistance (*ars*), and silver resistance (*sil*) genes and an ABC transport system.

## 3. Discussion

Our study demonstrates the complex molecular epidemiology of ESBL-producing *K. pneumoniae*, which are of growing concern for healthcare providers worldwide [3]. We found that each of the consecutive outbreaks in NICUs in three hospitals in a small geographic region was caused by a different dominant clone of *bla*_CTX-M_ -producing *K. pneumoniae.* In each NICU, along with the dominant clone, 1 to 3 secondary and satellite clones were also found. We determined that a satellite clone in NICU-2 was likely the origin of, or from a common reservoir as, the dominant clone in NICU-3, suggesting that the outbreaks did not represent completely independent nosocomial transmission within each institution. An epidemiological investigation focusing on common patients and healthcare workers between these two NICUs failed to uncover the link. We found no evidence of plasmid dissemination between clones as reported in other studies [6]. However, certain plasmids became more common because of clonal expansion of the ST35. Finally, the presence of singleton satellite clones D (in NICU-1) and K (in NICU-2) may represent ESBL importation from the community or from elsewhere in the hospital.

We found that all NICU-1 and most NICU-2 clones carried *bla*_CTX-M-15_. Interestingly, *bla*_CTX-M-15_ in NICU-1 was plasmid coded, while in the dominant clone of NICU-2 it was chromosomally coded. One NICU-2 satellite clone (ST35) carried *bla*_CTX-M-14_ and not *bla*_CTX-M-15_. The NICU-3-dominant clone, which likely originated from NICU-2, also carried *bla*_CTX-M-14,_ while the two satellite clones in NICU-3 carried *bla*_CTX-M-15_. In all isolates, the CTX-M genes were carried on the same transposable element, IS*Ecp1*, which is a common integrative element responsible for dissemination of ESBL genes. However, the flanking regions of the transposable elements varied between clones, suggesting multiple transposition events. We did not find evidence that such events linked the outbreaks investigated here, unlike previously reported events involving other transposable elements [13] or IS*Ecp1* [14].

Most ST types in our study are not frequently cited as ESBL-KP-causing outbreaks, except for ST15 [14,15], which appeared only as a satellite clone, and ST-37, a frequently found carbapenem- resistant strain [15]. Similarly, ESBL-KP clones responsible for other NICU outbreaks were not previously recognized as problematic nosocomial strains [6,7], which suggests that they were not part of the general hospital ecosystem.

We explored possible determinants of the dominant clones’ success. The dominant clones had on average a higher number of AMR genes, but no unique resistance mechanism. The dominant clones had various mechanisms of iron acquisition, including the citrate-mediated iron uptake system (*fec*), the *FhuDCB* locus, a chromosomal ABC-dependent Fe-siderophore uptake mechanism [16], and yersiniabactin. However, some satellite clones also carried these genes, including the plasmid-coded virulence factors. Thus, we did not find a specific determinant that explained the dominance of specific clones. We believe that in an era of wide dissemination of multidrug-resistant organisms, the success of a specific clone is likely a combination of particular clone characteristics, the NICU ecosystem, infection control practices, and chance.

Our study has several limitations. Among the potentially infected patients, only those hospitalized during the investigation were screened. Furthermore, not all isolates from ESBL-KP-positive patients were subjected to sequencing and only the selected dominant clones were sequenced using both long and short read sequencing technologies. However, using WGS on selected isolates provided useful information for understanding the ESBL-KP NICU outbreaks that could not be gleaned by standard epidemiology and traditional molecular techniques. Furthermore, by merging BOX-PCR with short and long read sequencing technologies, this study presented in-depth genomic analysis but at a lower cost. This approach may encourage smaller epidemiological laboratories to use genomic tools for epidemiological studies.

## 4. Materials and Methods

### 4.1. Patients and Setting

Three NICUs, within 20 km from each other, of different sizes were investigated in this study; NICUs 1 and 2 were of similar sizes with 16 and 24 beds, respectively. NICU-3 had 49 beds at the time of the outbreak. In all three hospitals, an index case of a neonate with bacteremia with an ESBL-producing *Enterobacteriaceae* isolated prompted screening of all exposed neonates. Identified carriers were placed in cohorts with isolation precautions and bacterial samples were sent to the reference laboratory at the National Center for Antibiotic Resistance and Infection Control.

### 4.2. Isolates and Molecular Analysis

Isolates were grown on MacConkey agar (Hy Laboratories Ltd., Rehovot, Israel) and identified to the species level by VITEK^®^ MS (bioMérieux SA, Marcy l’Etoile, France). Antibiotic susceptibility was determined by the VITEK^®^ 2 system. Confirmatory testing for ESBL production was carried out by double-disc diffusion according to CLSI guidelines. Isolates were stored at −80 °C and sub-cultured before testing. Clonality was determined using BOX-PCR [17]. Representative isolates from each BOX-PCR type and several replicates from the dominant clones were subjected to genomic analysis, to further understand the outbreaks’ genetic context. Only one isolate from the patient with multiple isolates was sent for sequencing. Isolate DNA was sent to the Sequencing Core at the University of Illinois at Chicago. Sequencing libraries were prepared using the Nextera XT kit (Illumina Inc., San Diego, CA, USA) and sequenced on the Illumina NextSeq 500 using the high output 2× 150 bp kit (Illumina Inc.). Each genome was sequenced to an average sequence depth of approximately 0.8 Gb per genome.

Quality validation of sequence data, adapter trimming, and ambiguous base removal were done by Fastp (v0.19.5) [18]. Next, samples were assembled using SPAdes (v3.12.0) [19] under default conditions. Gene calling and annotation were achieved using Prokka (v1.13.3) software [20]. One representative isolate (the isolate with the best N50 assembly score) from each NICU’s dominant clone was also sequenced using long read sequencing technology. Libraries were prepared using the multiplexing kit Rapid Barcoding Sequencing (SQK-RBK004) and sequenced on a MinION device (Oxford Nanopore Technologies (ONT), Oxford, UK) according to the manufacturer’s instructions. Sequence assembly and de-multiplexing were done using Guppy software (v3.2.2) (ONT). Illumina reads were quality screened using Fastp and then assembled together with ONT reads using Unicycler (v0.4.8) [21]. Annotation and downstream application were performed as described for the genomes sequenced only by Illumina technology.

### 4.3. Phylogenomics, Resistomes, and Virulence Factors

A list (Appendix A) of representative genomes of most KpI MLSTs [22] was downloaded from the SRA and PATRIC databases and was assembled and annotated in the same way as was done for this study’s isolates. All genomes (downloaded and new) were subjected to pan-genome analysis using Roary software [23]. Core genomes were aligned using default settings of the MAFFT (v7.407) alignment algorithm [24]. The resulting multi-gene alignment was subject to phylogenetic reconstruction using FastTree (v2.1.11) [25]. The resulting KpI tree was used to determine the phylogenetic relationship between clones in light of the greater KpI group in order to determine whether all clonal groups were distinct. The same strategy was employed for *K. pneumoniae* ST35 genomes listed in Appendix A. We used Kleborate (v0.3.0) (https://github.com/katholt/Kleborate) to identify and classify antibiotic resistance genes and MLST types and to predict the O-antigen and K-locus types.

### 4.4. Plasmid Content Analysis

Plasmid prediction was conducted by searching genome sequences against the PlasmidFinder database [26]. Plasmid MLST (pMLST) was assigned using the PubMLST web interface (https://pubmlst.org/plasmid/). For the genomes that were sequenced using long read technology and resulted in circular chromosomes and plasmids, the plasmid content was annotated and plotted using EasyFig v2.2.2 [27]. DNA fragments (5300–5800bp) from all sequenced isolates, with a transposable element carrying the *bla*_CTX-M-15_ gene were retrieved and plotted using EasyFig. These DNA sequences were compared to similar DNA fragments from genomics data previously published for *K. pneumoniae* strain ATCC BAA-2146 (Kpn2146) [10], CN1 [11] and UR033115 [12]. Contigs carrying *bla*_CTX-M-15_ were compared in a similar manner in order to examine the β-lactamase flanking regions.

All sequence data produced in this study were submitted to the sequence read archive (SRA) under bioproject accession number PRJNA529587 (Appendix A).

## 5. Conclusions

The investigation of three overlapping, consecutive ESBL-KP NICU outbreaks in nearby hospitals has shed light on their epidemiological complexity and the need to support investigations by WGS. We found that each outbreak had a dominant clone. Thus, even in the setting of high endemicity, a cluster of ESBL-KP should be treated as an outbreak and appropriately investigated.

## Figures and Tables

**Figure 1 antibiotics-09-00705-f001:**
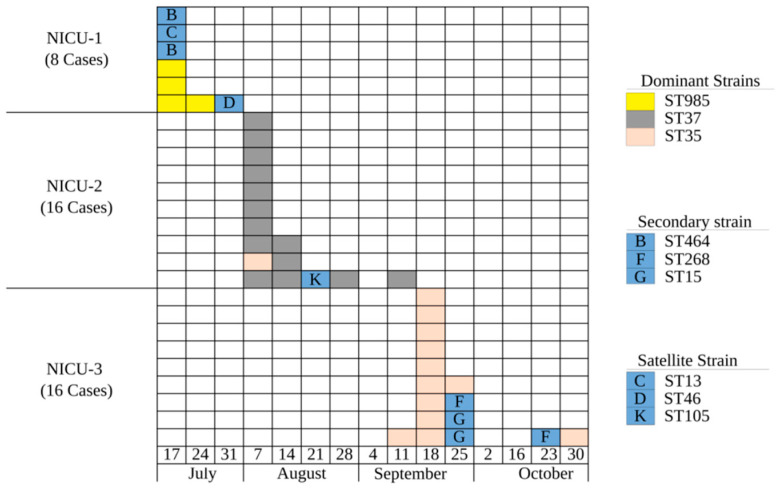
Epidemic curve of extended-spectrum beta-lactamase-producing *Klebsiella pneumoniae* (ESBL)-producing *K. pneumoniae* outbreaks in three neonatal intensive care units (NICUs): cases per week, 17 July–1 November 2017. Clones were determined by BOX-PCR and MLST and are color coded. Secondary strains (appeared more than once) and satellite strains (singletons) are coded by BOX-PCR letters as presented in Appendix A.

**Figure 2 antibiotics-09-00705-f002:**
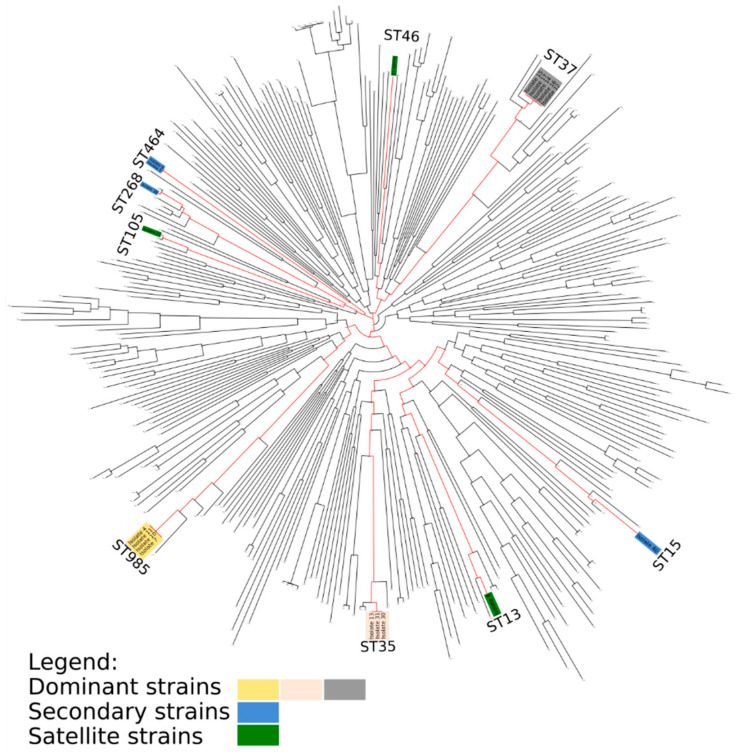
KpI, a major *K. pneumoniae* phylogenetic group, supplemented with ESBL-producing *K. pneumoniae* isolates from the neonatal intensive care unit outbreaks reported here. Dominant clones are in yellow, gray, and pink. Secondary and satellite strains are in blue and green, respectively.

**Figure 3 antibiotics-09-00705-f003:**
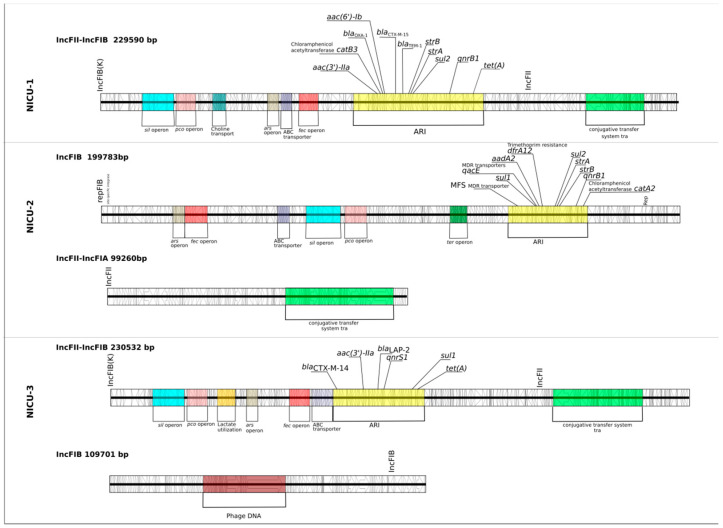
Schematic view of complete plasmids (over 10 Kbp) found by whole genome sequencing of dominant clones in outbreaks in three neonatal intensive care units. Antibiotic resistance genes and major operons are annotated.

**Figure 4 antibiotics-09-00705-f004:**
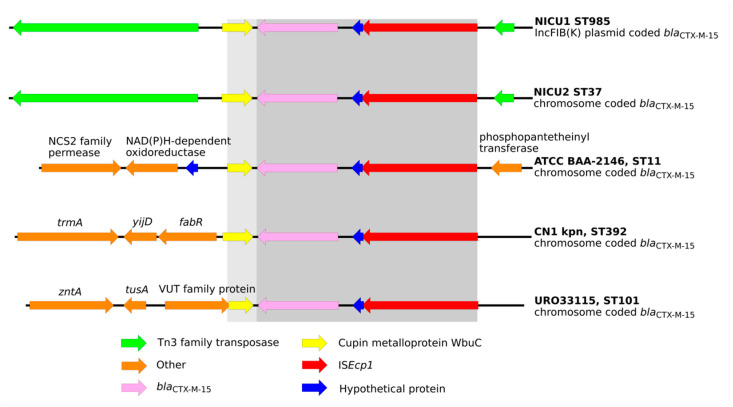
Schematic view of genomic fragments surrounding the IS*Ecp1* transposable element containing the bla_CTXM-15_ gene: comparison of fragments from the NICU-1- and NICU-2-dominant clones and previously described *K. pneumoniae* isolates. Dark gray indicates 100% similarity between all isolates and light gray indicates high similarity.

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
