# Peer review of "Investigation of Outbreaks of Extended-Spectrum Beta-Lactamase-Producing Klebsiella Pneumoniae in Three Neonatal Intensive Care Units Using Whole Genome Sequencing"

_antibiotics, 2020, doi:10.3390/antibiotics9100705_

Round 1

Reviewer 1 Report

Dear colleagues,

in this work you developped a molecular analysis on strains of Klebsiella pneumoniae recovered from three neonatal units.
You envisageated several questionning:
- "Was this a clonal outbreak involving all three hospitals?"
- "Were the outbreaks a result of plasmid dissemination between clones and NICUs?"
- "Were the outbreaks the result of dissemination of a blaCTX-M transposable element?"

Even though the study is quite complicated by itself, with numerous sample, several molecular sequencing and different kinds of bioinformatical analysis, the results are still very well explained and understandable.

I found interesting every attempt to unravel phenomenon of apparent causality.
In this case it appears that the 3 outbreaks can not be linked except for the timing of their occurence. To be more accurate, only one sub dominant strain was found in 2 of the 3 hospitals. That could be a marker of a common transmitter or reservoir ? Because of the clinical importance of K. pneumoniae, the extension of this work on the field of the pangenome study is quite interesting too. It seems useful to identified genes that could be involved in the dominancy of these pathogens.

suggestions:

the study seems not fully complete since all the children and strains recovered from them were not all included. Apparently some choices have been made (lack of time, money, organisation ? ) and several technics (vitek, box pcr, whole genome sequencing) where successively applied on the samples. I am not sure of the significance of the box pcr for the phylogeny of these strains, but the study still seems good because of the congruent results obtained with the other technics.

- Within the abstract L23 I would suggest "dominant strain" instead of "clone".
- a typing error occured L136-L139 with a remnant commentary.

Thank you very much for this interesting study and also for you honesty illustrated by your last paragraph highlightening the limitations of your work.

have a nice day

Author Response

Reviewer 1

Dear colleagues,

in this work you developped a molecular analysis on strains of Klebsiella pneumoniae recovered from three neonatal units. 
You envisageated several questionning:
- "Was this a clonal outbreak involving all three hospitals?"
- "Were the outbreaks a result of plasmid dissemination between clones and NICUs?"
- "Were the outbreaks the result of dissemination of a blaCTX-M transposable element?"

Even though the study is quite complicated by itself, with numerous sample, several molecular sequencing and different kinds of bioinformatical analysis, the results are still very well explained and understandable.

I found interesting every attempt to unravel phenomenon of apparent causality.
In this case it appears that the 3 outbreaks can not be linked except for the timing of their occurence. To be more accurate, only one sub dominant strain was found in 2 of the 3 hospitals. That could be a marker of a common transmitter or reservoir ? Because of the clinical importance of K. pneumoniae, the extension of this work on the field of the pangenome study is quite interesting too. It seems useful to identified genes that could be  

suggestions:

the study seems not fully complete since all the children and strains recovered from them were not all included. Apparently some choices have been made (lack of time, money, organisation ? ) and several technics (vitek, box pcr, whole genome sequencing) where successively applied on the samples. I am not sure of the significance of the box pcr for the phylogeny of these strains, but the study still seems good because of the congruent results obtained with the other technics.

Reply: BOX-PCR has been used to identify clonal relationships between isolates and is a well-documented method for this purpose (reference 14 in the manuscript). While this method is slowly getting outdated due to time investment and cost, it is still reliable.       

- Within the abstract L23 I would suggest "dominant strain" instead of "clone".

Reply: the term "dominant clone" was replaced with the term "dominant strain" in the abstract after the typing method was described (line 27).

- a typing error occured L136-L139 with a remnant commentary.

Reply: The commentary was removed an obvious error on our behalf. 

Thank you very much for this interesting study and also for you honesty illustrated by your last paragraph highlightening the limitations of your work.

have a nice day

Reply: thank you.

Reviewer 2 Report

A single comment, the authors wrote a carefully edited and original study. The results are presented in an original way, not the usual title, instead the authors ask a question, to which the reader is curious to know the answer.

Author Response

Reviewer 2:

A single comment, the authors wrote a carefully edited and original study. The results are presented in an original way, not the usual title, instead the authors ask a question, to which the reader is curious to know the answer.

Reply: Thank you for the kind remark

Reviewer 3 Report

The manuscript of Frenk et al. is analysing 43 isolates of ESBL producing Klebsiella pneumonia from 40 patients. The isolates are originating from three outbreaks in three different hospitals within 20 km away from each other. The manuscript has a potential, but I have some major concerns.

43 isolates from 40 patients: Does this mean that from three patients two isolates were obtained? How was this taken into account at statistics and selecting isolates for sequencing?

  1. Methods:
  • What kind of media was used for isolation of K. pneumonia?
  • Each genome was sequenced to an average of approximately 0.8G bp per genome. What exactly this means?
  • What does it mean »the phylogenetic relationship between clones in light of the greater KpI group«
  • How did you decide of which isolates the genome will be sequenced?
  • In table S1: which isolates are originating from each of the outbreak? What does it mean number 1 and 2? What does it mean dual technology? One of the box is empty, please fill it in. Which isolates are dominant clones and which not?
  1. The genomic tree is very small…difficult to read
  2. L74-75: Is this information really presented in Fig S1?
  3. L87-89: are the accession numbers of these genes and operons presented somewhere?
  4. Figure S1: What are meaning the characters after AR gene?
  5. L162-163: this statement needs more exact explanation.

English has to be checked throughout the manuscript.

In my opinion, the manuscript is written for someone who is professional in the subject ESBL producing K. pneumonia, what is ok, but on the other hand, the papers will attract more readers if some things would be explained better, such as what does it mean “KpI” phylogenetic tree and if clear legends would be added to all Tables and Figures.

Author Response

Reviewer 3:

Comments and Suggestions for Authors

The manuscript of Frenk et al. is analysing 43 isolates of ESBL producing Klebsiella pneumonia from 40 patients. The isolates are originating from three outbreaks in three different hospitals within 20 km away from each other. The manuscript has a potential, but I have some major concerns.

43 isolates from 40 patients: Does this mean that from three patients two isolates were obtained?

Reply: There was 1 patient with 4 isolates. We clarified this in the first paragraph of the Results section and in Supplementary Table 3 (table S1 in the revised manuscript).

How was this taken into account at statistics and selecting isolates for sequencing?

Reply: In line 226 we clarified that only 1 isolate from the patient with multiple isolates was sent for sequencing. Non of the analyses included this duplication. 

Reply:

  1. Methods:
  • What kind of media was used for isolation of K. pneumonia?

Reply: we added to the methods section: " Isolates were grown on MacConkey agar (Hy Laboratories Ltd, Rehovot, Israel) and identified to the species level by VITEK® MS (bioMérieux SA, Marcy l'Etoile, France) and antibiotic susceptibility was determined by the VITEK® 2 system."

  • Each genome was sequenced to an average of approximately 0.8G bp per genome. What exactly this means?

Reply: the depth of sequencing is determined by the amount of base pairs sequenced and in this case each genome was sequenced to the depth of an average of 0.8 Giga base pairs. We added to the text: "Each genome was sequenced to an average sequence depth of approximately 0.8Gb per genome."

  • What does it mean »the phylogenetic relationship between clones in light of the greater KpI group«

Reply: K. pneumonia was previously identified to consist of several phylogroups, as described by reference 19, one of which is the KpI. We used the reference genomes to show that all STs were phylogenetically distinguishable and are unrelated. We modified the text to specify this objective: "The resulting KpI tree was used to determine the phylogenetic relationship between clones in light of the greater KpI group in order to determine all clonal groups were distinct."

  • How did you decide of which isolates the genome will be sequenced?

Reply: the study's strategy was to minimize cost and complexity of analysis by selecting representative isolates from each BOX-PCR clonal group. Dominant clonal groups were sequenced using several isolates from which the best assembled isolates were subjected to long read sequencing. This methodology was addressed in lines 215-217 and 196-200. We added information on the selection of isolates for long read sequencing: " One representative isolate (which had the best N50 assembly score) from each NICU's dominant clone was also sequenced using long read sequencing technology."

  • In table S1: which isolates are originating from each of the outbreak? What does it mean number 1 and 2? What does it mean dual technology? One of the box is empty, please fill it in. Which isolates are dominant clones and which not?

Reply: The data in supplementary table 1 (table S2 in the revised manuscript) are complemented by Supplementary Table 3 (table S1 in the revised manuscript) and Figure 1. The ST of the isolates is described in Figure 1, which illustrates which are the dominant strains, he numbering is the isolates reference number which is used in table S3 to identify from which NICU, NCBI reference, assembly scores and more.

The dual technology refers to sequencing technologies and this was added to the text.  

The empty boxes indicate the absence of the named gene.

We added a hash (#) sign to represent isolates from dominant clonal groups.

  1. The genomic tree is very small…difficult to read

Reply: we increased the font of ST numbers for better reading.

  1. L74-75: Is this information really presented in Fig S1?

Reply: We added in line 77: " The ST35 NICU-2 isolate was indistinguishable from the ST35 NICU-3 isolates by core-genome alignment:  almost no dissimilarity distance was found between them (Fig. S1)." 

  1. L87-89: are the accession numbers of these genes and operons presented somewhere?

Reply: the accession numbers for the genomes can be found in table S1. However, the genes are available only within the genebank annotation. For example, the fec operon genes have the locus tags ranging from E4W57_025790 to E4W57_025820 on genomic fragment CP059380.1 in biosample SAMN11270019. Figure 3 would be impossible to read if we need to add these accession numbers for all the genes in all the genomes. However, the data is freely available in public repositories for anyone interested replicating the analysis.  

  1. Figure S1: What are meaning the characters after AR gene?

Reply: I assume the reviewer meant figure S3, the names of the genes and their indication in the illustration are written after the "AR gene". The names were written in "popular" form and we fixed this to follow scientific gene form.

  1. L162-163: this statement needs more exact explanation.

Reply: we reworded the sentence: "In each NICU, along with the dominant clone, 1-3 secondary and satellite clones were also found."

English has to be checked throughout the manuscript.

Reply: the manuscript was edited 

In my opinion, the manuscript is written for someone who is professional in the subject ESBL producing K. pneumonia, what is ok, but on the other hand, the papers will attract more readers if some things would be explained better, such as what does it mean “KpI” phylogenetic tree and if clear legends would be added to all Tables and Figures.

Reply: We added significant information to the legends of each figure for the ease of reading

We would like to thank the reviewer for the thorough review and the many constructive comments that improved this manuscript.

Reviewer 4 Report

Authors used whole genome sequencing (WGS) to investigate the epidemiology of consecutive and overlapping outbreaks caused by ESBL-KP in NICUs in three hospitals in close proximity. A dominant clone was found in each outbreak. Authors concluded that the overlapping ESBL-KP outbreaks were a combination of clonal transmission within NICUs, possible transposable element transmission between NICUs, and repeated importation of ESBL-KP from the community.

Comment 1: The manuscript presents a very interesting topic of the epidemiological complexity but could addresses the significance and innovations better.

Comment 2: ESBL-KP has become a raising threat to nosocomial outbreak and community spread. Is there a reason why the authors choose to focus on NICU? Were there outbreaks inside each of the three hospitals?

Comment 3: Result 2.4. What may have contributed to the success of the dominant clones? This seems to be an incomplete paragraph (line 136-138). However, this is an extremely interesting discussion and might be potential pharmacological targets and/or key of infection control. More experimental data could be included. However, only the selected dominant clones were fully sequenced. Limited conclusions could be drawn.

Author Response

Reviewer 4:

Authors used whole genome sequencing (WGS) to investigate the epidemiology of consecutive and overlapping outbreaks caused by ESBL-KP in NICUs in three hospitals in close proximity. A dominant clone was found in each outbreak. Authors concluded that the overlapping ESBL-KP outbreaks were a combination of clonal transmission within NICUs, possible transposable element transmission between NICUs, and repeated importation of ESBL-KP from the community.

Comment 1: The manuscript presents a very interesting topic of the epidemiological complexity but could addresses the significance and innovations better.

Reply: we added a few words addressing this study's methodological innovation and importance at the end of the discussion: "Furthermore, by merging BOX-PCR with short and long read sequencing technologies, this study presented in-depth genomic analysis but at a lower cost. This approach may encourage smaller epidemiological laboratories to use genomic tools for epidemiological studies." 

Furthermore, important points of significance are included in the text such as the discovery of a new isolate with a blaCTX-M-15 and the wide spread of ISEcp1.

Comment 2: ESBL-KP has become a raising threat to nosocomial outbreak and community spread. Is there a reason why the authors choose to focus on NICU?

Reply: The outbreaks occurred in neonatal intensive care units and therefore focused on them. 

Were there outbreaks inside each of the three hospitals?

Reply: yes. This is explained in lines 55-57 and 213-216 and in Figure 1.

Comment 3: Result 2.4. What may have contributed to the success of the dominant clones? This seems to be an incomplete paragraph (line 136-138). However, this is an extremely interesting discussion and might be potential pharmacological targets and/or key of infection control. More experimental data could be included. However, only the selected dominant clones were fully sequenced. Limited conclusions could be drawn.

Reply: all the isolates were carbapenem susceptible as can be seen in the complete antibiotic resistance profile presented in table S1. This hopefully addresses the reviewers concerns for a complete search for potential pharmacological targets.

We would like to thank the reviewer for the thorough review and  constructive comments that improved this manuscript.

Reviewer 5 Report

Frenk et al. have studied in detail outbreaks of extended-spectrum beta-lactamase-producing Klebsiella pneumoniae in three neonatal intensive care units using whole genome sequencing. The manuscript is well written, and experiment have been well designed. I have some minor questions stated below

  1. What were the age groups of 40 patients taken? Although its neonatal intensive care units but was sample from term neonates or preterm or were taken in random?
  2. What sample was taken for whole genome sequencing?

Author Response

Reviewer 5

Frenk et al. have studied in detail outbreaks of extended-spectrum beta-lactamase-producing Klebsiella pneumoniae in three neonatal intensive care units using whole genome sequencing. The manuscript is well written, and experiment have been well designed. I have some minor questions stated below

  1. What were the age groups of 40 patients taken? Although its neonatal intensive care units but was sample from term neonates or preterm or were taken in random?

Reply: All neonates in the ward were sampled once the outbreak was identified, thus, all ages were included.  

  1. What sample was taken for whole genome sequencing?

Reply: In line 222 we explain "Representative isolates from each BOX-PCR type and several replicates from the dominant clones were subjected to genomic analysis , to further understand the outbreaks' genetic context. "

In line 235 we added information on the selection of isolates for long read sequencing: "One representative isolate (which had the best N50 assembly score) from each NICU's dominant clone was also sequenced using long read sequencing technology."

We would like to thank the reviewer for the constructive input.

Round 2

Reviewer 3 Report

The manuscript has been corrected accordingly.

Author Response

no changes have been made

Reviewer 4 Report

The manuscript introduced interesting in-depth genomic analysis in ESBL-KP outbreaks in NICUs in three hospitals in close proximity. The experiments have been well designed. The authors presented results in a way to address several commonly asked questions which also successfully elicits readers' curiosity. The edited version is easier to read and obvious errors were edited.

Author Response

no changes have been made